# Alternation emerges as a multi-modal strategy for turbulent odor navigation

Nicola Rigolli[1,2,3†], Gautam Reddy[4,5,6†], Agnese Seminara[1,2]*, Massimo Vergassola[7]*

[1]MalGa, Department of Civil, Chemical and Mechanical Engineering, University of Genova, Genova, Italy; [2]Institut de Physique de Nice, Université Côte d'Azur, Centre National de la Recherche Scientifique, Nice, France; [3]Department of Physics and INFN Genova, University of Genova, Genova, Italy; [4]NSF-Simons Center for Mathematical and Statistical Analysis of Biology, Harvard University, Cambridge, United States; [5]Physics & Informatics Laboratories, NTT Research, Inc, Sunnyvale, United States; [6]Center for Brain Science, Harvard University, Cambridge, United States; [7]Laboratoire de physique de l'École Normale Supérieure, CNRS, PSL Research University, Sorbonne Université, Paris, France

**Abstract** Foraging mammals exhibit a familiar yet poorly characterized phenomenon, 'alternation', a pause to sniff in the air preceded by the animal rearing on its hind legs or raising its head. Rodents spontaneously alternate in the presence of airflow, suggesting that alternation serves an important role during plume-tracking. To test this hypothesis, we combine fully resolved simulations of turbulent odor transport and Bellman optimization methods for decision-making under partial observability. We show that an agent trained to minimize search time in a realistic odor plume exhibits extensive alternation together with the characteristic cast-and-surge behavior observed in insects. Alternation is linked with casting and occurs more frequently far downwind of the source, where the likelihood of detecting airborne cues is higher relative to ground cues. Casting and alternation emerge as complementary tools for effective exploration with sparse cues. A model based on marginal value theory captures the interplay between casting, surging, and alternation.

*For correspondence:
agnese.seminara@unige.it (AS);
massimo.vergassola@phys.ens.
fr (MV)

†These authors contributed equally to this work

## Editor's evaluation

This work demonstrates how animals can combine different kinds of sensing actions to improve the accuracy of finding the sources of olfactory signals. Previous work have provided an explanation of the casting and surging behaviors used to find the plume in order to navigate towards the sources. This work adds "alternation" - sniffing in the air far from the ground - that animals can do by rearing on hind legs. The authors show that alternation occurs more frequently far away from the source and that this can be explained by the marginal value theory.

## Introduction

The behavior of dogs alternating between sniffing in the air and close to the ground while tracking an odor scent is familiar to any cynophilist (*Thesen et al., 1993*; *Steen et al., 1996*; *Hepper and Wells, 2005*; *Jinn et al., 2020*). A similar behavior is well documented for rodents, where the slowdown associated with sniffing in the air can lead to stopping and rearing of the animal on its hind legs (*Khan et al., 2012*; *Gire et al., 2016*). This 'alternation' between the two sensorimotor modalities strongly suggests that both airborne and ground odor cues may be exploited by animals and integrated into a multi-modal navigation strategy.

Despite the behavior's familiarity, the reasons underlying the alternation between airborne and ground odor cues as well as the rationale of their integration are largely unknown (*Reddy et al., 2022*). Rodents may rear on their hind legs for a variety of reasons, generally associated with novelty detection, information gathering, anxiety and fear, as reviewed by *Lever et al., 2006*. In the laboratory odor-guided search developed by *Gire et al., 2016*, mice tend to pause and rear more often in the early stages of the task. This empirical observation is consistent with rearing in response to novelty and the hypothesis that raising their head may provide the animals additional olfactory information (*Lever et al., 2006*). On the physical side, it is expected that ground and airborne odor signals convey complementary information even if both signals are generated by a single source of odors. Indeed, airborne odors are valuable as distal cues because they are transported rapidly over long distances by flows that are often turbulent. The downside of airborne cues is that turbulence breaks odor plumes in discrete pockets, which can only be detected sparsely (*Murlis and Jones, 1981*; *Shraiman and Siggia, 2000*; *Falkovich et al., 2001*; *Celani et al., 2014*). Furthermore, since local gradients are randomized in relation to the source direction at the timescales of olfactory searches, gradient-ascent navigation strategies are not possible (*Vergassola et al., 2007*). Conversely, odor cues close to the ground are smoother and more continuous than odors in the air (*Fackrell and Robins, 1982*; *Nironi et al., 2015*). The physical reason is that viscous effects make fluids slow down while flowing close to the ground at rest. As a result, boundary layers are created and the structure of the flow depends on the height from the ground (*Anderson, 2005*). In short, airborne cues are more sparse and difficult to exploit for navigation than ground signals, yet they are faster and cover longer ranges. It is therefore likely that the relative value of sniffing closer vs. farther from the ground depends on the position of the searcher relative to the source via the statistics of odor detections that the searcher experiences. The corresponding decision of the most appropriate sensorimotor modality in response to a given history of detections is then expected to play a major role in determining an effective navigational strategy.

Here, we propose a normative theory to rationalize alternation behavior and the integration of airborne and ground-based olfactory modalities. First, we create a well-controlled setup using fully resolved numerical simulations of the odor concentration field generated by an odor source in a channel flow. Simulations produce realistic odor plumes over distances of several meters to the source. Second, we ask what is the optimal strategy to reach the olfactory source (target) as identified by machine learning methods. Specifically, we formalize the olfactory search problem as a partially observable Markov decision process (POMDP) and use state-of-the-art methods to solve the corresponding Bellman optimization problem. The agent performing the olfactory search is given the choice between the actions of freely moving while sniffing on the ground or stopping and sniffing in

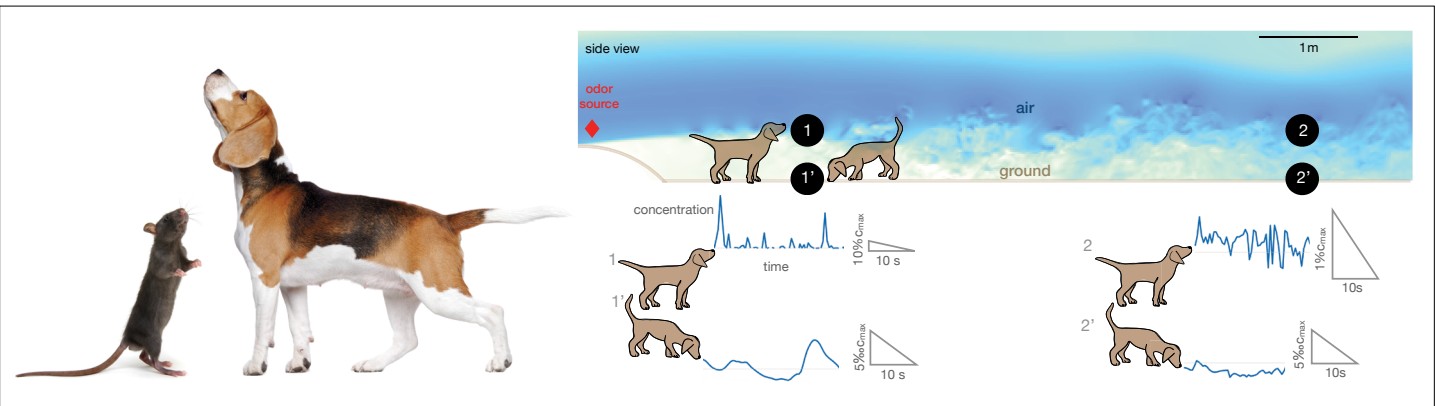

**Figure 1.** Alternation between different olfactory modalities is widespread in animal behavior. Left: A rodent rearing on hind legs and smelling with its nose high up in the air; a dog performing a similar behavior. Credit: irin-k/Shutterstock.com and Kasefoto/Shutterstock.com. Right: Side view of the direct numerical simulation of odor transport. Shades of blue give a qualitative view of the intensity of velocity fluctuations in a snapshot of the field. Colors are meant to emphasize the boundary layer near the bottom, where the velocity is reduced by the no-slip condition at the ground. Representative time courses of intense intermittent odor cues in air (sampled at 53 cm from the ground, locations marked with 1 and 2) vs. smoother and dimmer cues near the ground (sampled at 5 mm from from the ground, locations marked with 1' and 2'). Different animals sniff at different heights, which alters details of the plumes but does not affect the general conclusions. Data obtained from direct numerical simulations of odor transport as described in the text, see Materials and methods for details.

the air. Solving the POMDP yields a policy of actions taken in response to a history of odor stimuli, which is encoded into a set of probabilistic beliefs about the location of the source. While the searcher could a priori reach the target using ground cues only, we demonstrate that learned strategies generically feature alternation between airborne and ground odor cues. Alternation is more frequent far downwind of the source and is associated with casting. The emergence of this non-trivial behavior is rationalized as the need to gather information under strong uncertainty from distal airborne cues, which leads to better long-term reward compared to local exploration for the source or proximal ground cues.

## Model

Consider a food source located outdoors which exudes odor at a constant rate. The odor is steadily carried by the wind and dispersed due to turbulent fluctuations. In the atmosphere, turbulent transport of odors dominates molecular diffusion and determines the statistics of the odor signal. A plume-tracking agent which enters the area downwind has to navigate its way upwind toward the source by sniffing the ground or pausing to sniff in the air for odor.

The statistics of odors on the ground is profoundly different from the statistics of odors in the air (see representative time courses in *Figure 1*). In the situation represented in *Figure 1*, the divide between air and ground is dictated by the fluid dynamics in the boundary layer close to the ground. In our direct numerical simulations (DNS) of odor transport, the air travels in a channel from left and hits an obstacle at 25 cm/s, which generates turbulence. The simulations are designed to resolve the dynamics at all relevant scales, from few mm to several m, which demands massive computational resources (see Materials and methods for details of the numerical scheme). Odor is released from a spherical source of size 4 cm located 56 cm above the ground. At the height of the source, odor is efficiently carried several meters downwind within pockets of odor-laden air which remain relatively concentrated, but are distorted and broken by turbulence. Thus, odor in the air is intense but intermittent, that is, it varies abruptly in time. Conversely, odor near the ground is smoother but also less intense (see *Figure 1*). It is smoother because the air in contact with the ground is still, which creates a nearly stagnant boundary layer where the disruptive effect of turbulence is tamed; it is less intense because odorant molecules generally bind to surfaces, which act as odor sinks (see comprehensive discussions in the context of the design of olfactory tasks *Gorur-Shandilya et al., 2019*).

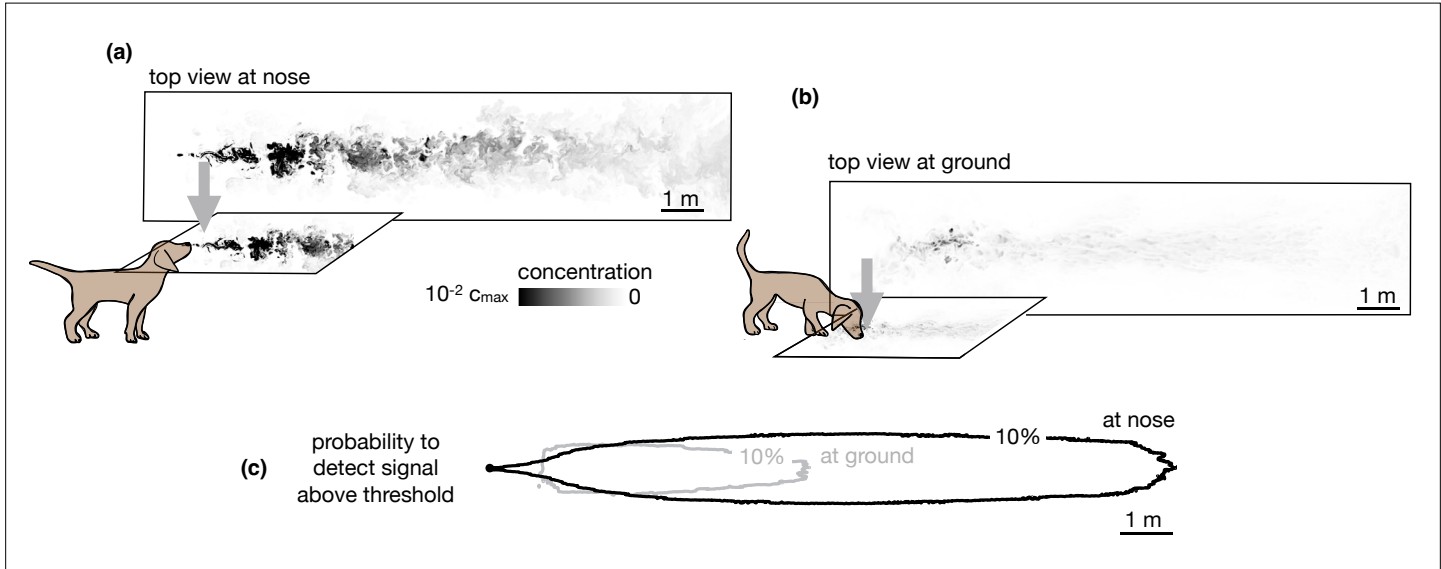

**Figure 2.** Snapshots of odor plume obtained from direct numerical simulations of the Navier-Stokes equations in three spatial dimensions. Top view of the odor plume (**a**) at nose height and (**b**) at ground. (**c**) 10% isoline of the probability to detect the odor $r(x, y)$ (defined as the probability that odor is above a fixed threshold of 0.14% with respect to the maximum concentration at the source) at the ground (gray) and at the nose height (black). Data to generate *Figure 2* are public on Zenodo (https://zenodo.org/record/6538177#.Yqrl_5BByJE).

The online version of this article includes the following figure supplement(s) for figure 2:

**Figure supplement 1.** Detection probability maps in the air and at the ground.

Our simulations specifically consider the limiting case of total adsorption of odor molecules. Qualitatively, similar results are expected for models intermediate between total adsorption and total reflection, where particles have a finite likelihood of being adsorbed or re-emitted in the bulk. Total (or partial) depletion of odors at the ground surface implies that an agent with a finite detection threshold can only sense ground odors near the source. Conversely, the agent can sense larger plumes in the air than on the ground compare top views in the air vs. the ground in *Figure 2a-b*, and it is able to detect odor in air across a more extended area (see *Figure 2c* and *Figure 2—figure supplement 1*).

The statistics of odor encounters detected along the search path of a plume-tracking agent provides useful information about the location of the source, which guides subsequent navigation. We consider an agent moving along a path $r_1, r_2, \ldots, r_t$ while measuring the odor signal $o_1, o_2, \ldots, o_t$. The agent's present knowledge is fully summarized by the posterior distribution of the agent's location relative to the source, $\mathbf{b}_t$, also called the belief vector. The agent computes Bayesian updates of the belief $\mathbf{b}_t$ using a model (the likelihood of odor detections) and the current observation $o_t$. At each time step, $t_s$, the agent decides among six alternatives: (i–iv) move to one of the four neighboring locations while sniffing the ground, (v) stay at the same location and sniff the ground, or (vi) stay at the same location and sniff the air. The agent decides among these choices based on the long-term reward it expects to receive, as discussed in the next paragraph.

We pose the agent's task in the framework of optimal decision-making under uncertainty. The agent's actions are driven by a unit reward received when it successfully finds the source. Rewards are discounted at a rate $\lambda$, that is, the expected long-term reward is $\langle e^{-\lambda T} \rangle_T$. Here, $T$ is the time taken to find the source and the expectation is over the prior knowledge available to the agent, its navigational strategy, and the statistics of odor encounters. The expected long-term reward or *value*, $V(\mathbf{b}_t)$, given the current state of knowledge, $\mathbf{b}_t$, can be calculated using the Bellman equation, a dynamic programming equation which takes into account all possible future trajectories of an optimal agent. Specifically, we obtain the Bellman equation (see Materials and methods for details)

$$V\left(\mathbf{b}_t\right) = \max_a \left\{ \Gamma_a + \gamma(1 - \Gamma_a) \sum_{o_{t+1}} P(o_{t+1}|\mathbf{b}_t, a) V\left(\mathbf{b}_{t+1}\right) \right\}, \tag{1}$$

where $\Gamma_a$ is the probability of finding the source immediately after taking action $a$, $\gamma \equiv e^{-\lambda t_s}$ and the probability $P(o_{t+1}|\mathbf{b}_t, a)$ of observing $o_{t+1}$ is determined by the physical environment and the signal detection threshold of the agent. Intuitively, the terms in the argument of the max function in *Equation 1* represent the value of finding the source, detecting the odor signal or not detecting the odor signal, each event being weighted by its probability. The optimal action is the one that maximizes the value, that is, the parenthesis on the right-hand side of *Equation 1*. For simplicity, we discretize observations into detections (odor signal above a fixed threshold) and non-detections, which implies that the behavior depends solely on the probability per unit time of detecting the odor on the ground or in the air (*Figure 2c*). Thus, the agent uses a (partially inaccurate) Poissonian detection model. The model is partially inaccurate as the average detection rate does match the simulated odor plumes but the model of the agent lacks the appropriate spatiotemporal correlations because detections are independent in the Poissonian model and they are not in the real flow. See Results for more details and the corresponding performance.

The decision-making dynamics form a POMDP (see Materials and methods for a brief introduction to POMDPs). To solve *Equation 1*, we use approximate methods, which exploit a piecewise linear representation of the value function: $V(\mathbf{b}_t) \approx \max_i \{\alpha_i.\mathbf{b}_t\}$ (*Shani et al., 2013*). The $\alpha_i$'s are a set of hyperplanes, which are found by simulating trajectories of the agent along exploratory search paths (Materials and methods). Given an existing set of $\alpha_i$'s, the rule for adding new hyperplanes is obtained by plugging in $V(\mathbf{b}_t) \approx \max_i \{\alpha_i.\mathbf{b}_t\}$ into *Equation 1*. As training progresses, the set of hyperplanes grows and yields increasingly accurate approximations of the value function. For each test run, we begin with a uniform prior distribution and simulate the POMDP until the agent finds the source. If the agent does not find the source within 1000 steps, we interrupt the simulation. The time step, $t_s$, and the distance traveled at each step are set such that the agent sniffs three times per second and at every step it moves 12 cm.

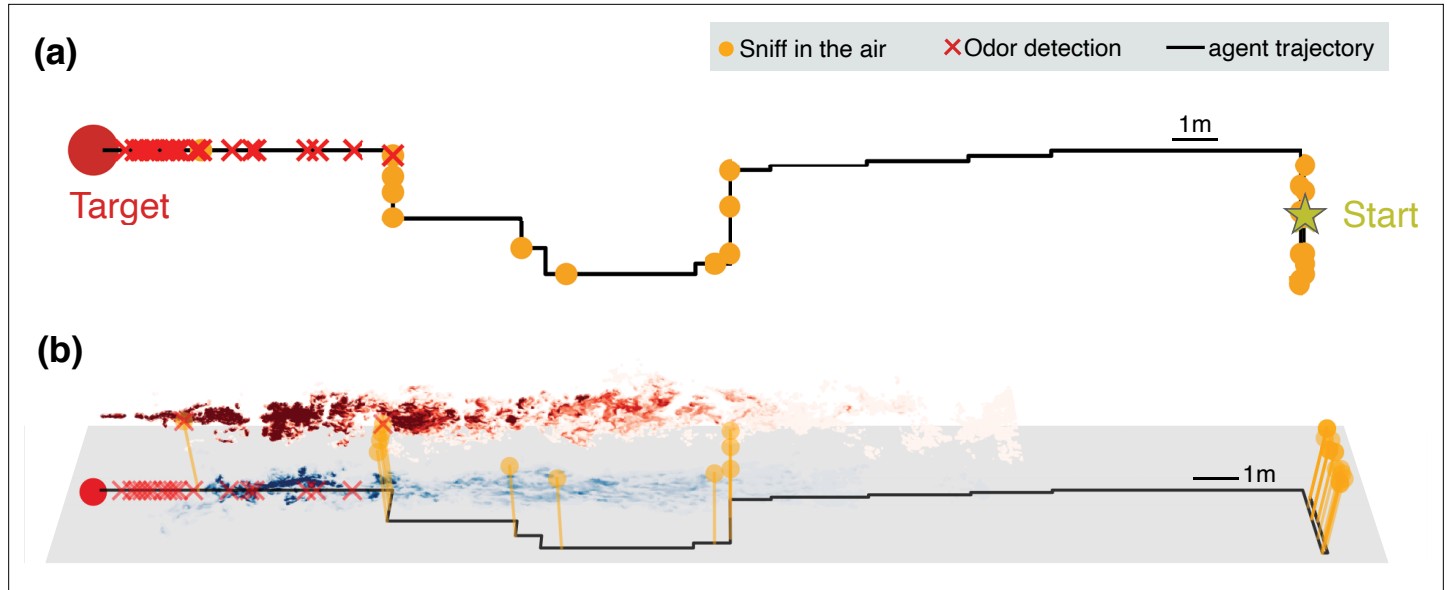

**Figure 3.** Representative trajectories undertaken by an agent learning how to reach the source of a turbulent odor cue. (**a**) Top view of a representative trajectory at the end of training. (**b**) Three-dimensional view of sample trajectory from panel (**a**), superimposed to two snapshots of odor plumes near ground (shades of blue) and in the air (shades of red). Trajectories are obtained by training a partially observable Markov decision process (POMDP), where the agent computes Bayesian updates of the belief using observations (odor detection or no detection) and their likelihood (detection rates from simulations of odor transport). Agents trained with this idealized model of odor plumes successfully track targets when tested in realistic conditions (see *Figure 3—video 1*).

The online version of this article includes the following video and figure supplement(s) for figure 3:

**Figure supplement 1.** Training efficiency and computational cost of partially observable Markov decision process (POMDP).

**Figure 3—video 1.** Trajectory of a partially observable Markov decision process (POMDP) agent navigating the realistic odor plumes simulated numerically.

https://elifesciences.org/articles/76989/figures#fig3video1

## Results

### The agent navigates by alternating between sniffing the ground and air

An agent initially downwind of the odor source learns to navigate the odor plume to maximize the discounted reward described previously, that is, to minimize the search time. The upshot of the learning phase is that the final search policy alternates between sniffing on the ground and the air (*Figure 3*). The average time taken to reach the source reduces considerably with the training time, indicating the emergence of an effective navigational strategy (*Figure 3—figure supplement 1A, B*).

The trajectories learnt by the agent display a variety of behaviors reminiscent of those exhibited by animals, which include wide crosswind casts interleaved with upwind surges. Notably, the agent exhibits a recurring motif which cycles between moving to a new location and pausing to sniff in the air. The alternating behavior emerges directly as a consequence of the statistics of the physical environment in spite of pausing to sniff in the air, which leads to the cost of a stronger discount in the reward.

When, where, and why does the agent sniff in the air? Trajectories shown in *Figure 3* exhibit extensive alternation at the beginning of the search when the agent is far downwind compared to when it is close to the source. A quantitative analysis across training and test realizations confirms that the agent's rate of sniffing in the air is significantly higher farther away from the source (*Figure 4(a)*). This observation is rationalized by the greater probability of detecting an odor signal in the air at distant locations (*Figure 2(c)*) despite the increased intermittency in the airborne signal (*Figure 1*). In spite of the added cost entailed by slowing down locomotion, sniffing in the air ultimately speeds up the localization of the source (*Figure 4(b)*). This behavior is maintained across different training realizations and when the discount factor, $\gamma$, is reduced so that the delay incurs a greater cost. In sum, alternation

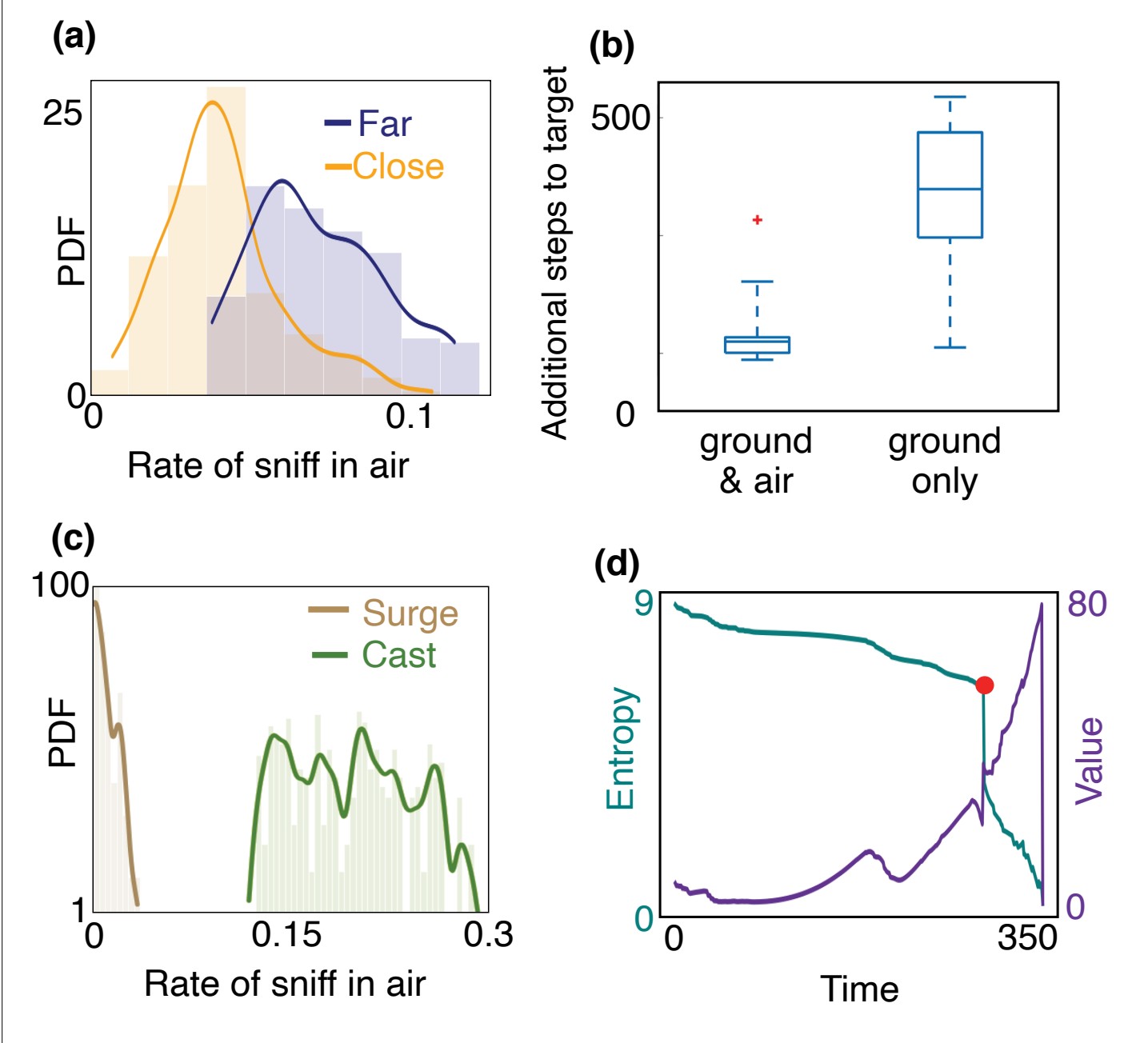

**Figure 4.** Empirical characterization of the alternation between olfactory sensory modalities. (**a**) The agent sniffs more often in the air when it is far from the source, that is, outside of the airborne plume. The rate of sniffing in the air is the fraction of times the agent decides to sniff in the air rather than move and sniff on the ground. The fraction is computed over the entire trajectory in the conditions identified in the different panels. Statistics is collected over different realizations of the training process and many trajectories, with different starting positions (see Materials and methods for details). (**b**) The number of steps needed to reach the target minus the number of steps needed to travel from the starting position to the source in a straight line. The horizontal line marks the median, boxes mark 25th and 75th percentiles; red dot: outlier (value exceeds 75th percentile + ×1.5 interquartile range). Dashed lines mark 10th and 90th percentile. For reference, a straight line from the center of the belief to the source is 240 steps. Agents that are given the possibility to pause and sniff in the air are able to reach the target sooner than agents that can only sniff on the ground. (**c**) Agents sniff in the air once every five steps on average when they cast (three consecutive steps crosswind), whereas they only sniff in the air once every 60 steps while surging upwind (three consecutive steps upwind). (**d**) Entropy (cyan) and value (purple) of the belief vs. time, along the course of one trajectory. The red dot indicates a detection, which provides considerable information about source location and thus makes entropy plummet and value increase.

The online version of this article includes the following figure supplement(s) for figure 4:

*Figure 4 continued on next page*

*Figure 4 continued*

**Figure supplement 1.** Empirical characterization of the alternation between olfactory sensory modalities when $\gamma = 0.95$.

**Figure supplement 2.** Performance of an agent in a different environment during test.

emerges as a robust, functional aspect of an effective long-term strategy of olfactory search (see also *Figure 4—figure supplement 1*).

A striking feature of the trajectories in *Figure 3* is the strong correlation between casting and sniffing the air before the first detection is made. To quantify this effect, we categorize the agent's behavior into casts (persistent crosswind movements) and surges (upwind movement), and measure the rate of sniffing the air (fraction of time spent executing action (vi) – staying at the same location and sniffing the air – among the six possible actions offered to the searcher) for both of these behaviors. We consider the agent to be surging if it moves $k$ consecutive steps upwind and casting if it moves $k$ consecutive steps crosswind or sniffs in the air. We use $k = 3$, results shown hereafter do not depend strongly on this choice. We find that the rate of sniffing in the air is typically an order of magnitude greater during casts as compared to surges (*Figure 4(c)*), indicating that alternation is tightly linked to the switch between casting and surging. Casting has been classically interpreted as a strategy for efficient exploration in an intermittent environment. The coupling between casting and alternation observed here suggests that sniffing in the air is an alternative mode of exploration which aids and complements casting when searching for a sparse cue. Exploration dominates the first part of the search until the first detections which substantially reduce uncertainty (see entropy of the posterior distribution in *Figure 4(d)*).

Overall, we led to the following picture of the search dynamics. At the beginning of the search, the agent has a broad prior that is much larger than the odor plume of size $\sim x_{\mathrm{thr}} \times y_{\mathrm{thr}}$, where $x_{\mathrm{thr}}$ is the plume length and $y_{\mathrm{thr}}$ is the plume width when sniffing in the air. The agent then has to identify and home into the $x_{\mathrm{thr}} \times y_{\mathrm{thr}}$ region that contains the odor plume. The bottleneck in this phase is the scarcity of odor detections, which require an efficient exploration strategy. Once the odor plume is detected, the agent knows it is near the source and the search is driven by surface-borne odor cues, while the frequency of sniffing in the air is significantly reduced. In short, our simulations show that the behavior can be split into two distinct phases: (1) an initial exploration phase accompanied by extensive casting and alternation, where the agent attempts to localize the plume, and (2) odor-guided behavior in a regime relatively rich in cues, which enable the agent to precisely locate the source within the plume.

We conclude this section noting that the above remarks are expected to hold more generally than in the specific setup of our simulations. *Figure 3—video 1* shows that the same behaviors are displayed by agents navigating a realistic plume despite their learning in a (partially inaccurate) Poissonian model of odor detections. This finding indicates the robustness of the learning scheme to inaccuracies in the model of the environment, which are inevitably present in any realistic situation. More specifically, the static information provided by the average detection rate map is found to be sufficient for navigation and alternation. While more information on dynamical spatiotemporal correlations may help further improve performance, the fundamental requirement for alternation is the presence of wider detection rate maps in the air than on the ground. Thus, as long as this feature is preserved, we expect agents to display alternating behaviors and the two phases mentioned above. In particular, these properties should hold also when different models of odor transport are employed by the searcher and/or surface adsorption chemistry is more involved than pure adsorption. *Figure 4—figure supplement 2* shows that the agent navigates successfully when the odor statistics is different from the one that was used for training, suggesting that the strategy is robust to a broad set of different flow conditions.

## Intuition

We now proceed to understand the transition between the two distinct phases of search that were identified previously. While a detailed analysis of each individual decision is challenging, we can gain intuition by decomposing the agent's overall behavior into segments. Each segment is then rationalized by examining how the agent explores the locations where it believes it can find an odor signal or

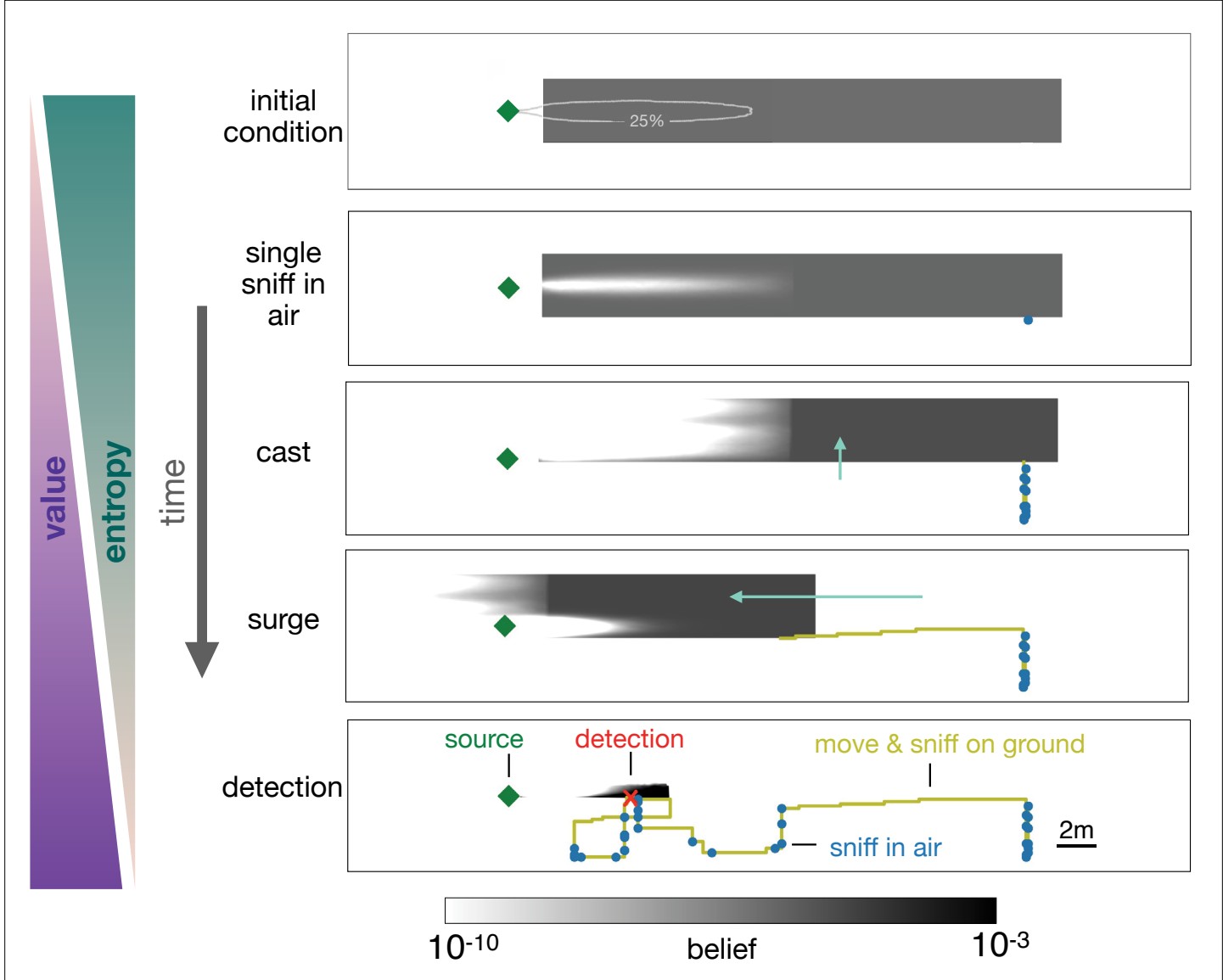

**Figure 5.** Progression of the belief the agent has about its own position relative to the source. From top, first panel: Before starting the search the agent has a flat belief about its own position, much broader than the plume in air represented by the 25% isoline of the probability of detection. Second: Belief after a single sniff in the air and no detection. The white region corresponds to the extent of the plume in air and indicates that because the agent did not detect the odor, it now believes it is *not* within the plume right downstream of the source. Third: As the agent casts, its belief about its own position translates sideways with it; additionally, at each sniff in the air with no detection, the belief gets depleted right downstream of the source, as in the panel right above. As a result, the cast-and-sniff cycle sweeps away a region of the belief as wide as the cast and as long as the plume. Fourth: As the agent surges upwind, its belief about its own position translates forward with it; additionally, as it sniffs on the ground with no detection, the belief gets depleted in a small region right downstream of the source, corresponding to the extent of the plume on the ground. Fifth: After detection, the belief shrinks to a narrow region around the actual position of the agent, which leads to the final phase of the search within the plume. Green (purple) wedges indicate that the entropy of the belief decreases (value of the belief increases) as the agent narrows down its possible positions (and approaches the source).

the source. For this purpose, let us examine how the agent's belief of its location relative to the source evolves as the search proceeds.

In the representative example depicted in *Figure 5*, the agent begins with a uniform prior belief, much larger than the plume, as shown in the top row of *Figure 5*. The agent makes its first action by sniffing in the air and does not detect an odor signal. Since odor is not detected, the likelihood that the agent is immediately downwind of the source is reduced, which leads to a posterior belief updated via Bayes' rule (second row, *Figure 5*). The agent proceeds by casting crosswind in a loop

while occasionally pausing to sniff in the air (third row, *Figure 5*), after which it executes an upwind surge (fourth row, *Figure 5*). The decision to surge at that specific moment can be understood from examining the belief immediately before the surge: because the agent did not detect any odor over the entire cast-and-sniff sequence, the likelihood that the agent is located near the source, that is, within the plume, is extremely low (third row, *Figure 5*).

At this point, it is more valuable to surge upwind rather than continuing to explore the same area. By surging forward, the agent is now more likely to encounter the plume, which enables it to effectively explore the remaining part of the belief. The key to the above argument is that the agent lacks knowledge of its position relative to the source, and it acts so as to narrow down its belief. Indeed, over the course of a search, entropy of the belief steadily declines, while its value increases (*Figure 4(d)*).

A repetition of the sequence of casting, alternation, and surging follows as the agent steadily narrows down the belief, until it finally detects the odor (bottom panel, *Figure 5*). The detection shrinks the posterior to a small patch which makes entropy plummet (*Figure 4(d)*) and leads the agent rapidly to the source. The first detection event (identified by the red dot in *Figure 4(d)*) is what marks the transition between searching *for* the plume and searching *within* the plume, as discussed above.

## Searching for airborne cues

We now expand on the intuition above by introducing a simplified, quantitative model of the search. We aim to address the search dynamics in the initial phase before detection, when the agent searches for the plume. This is the key phase as the localization of the plume largely dominates the search time (see red dot in *Figure 4(d)*). We identify three main questions about the search, which we address in more detail below: (1) how wide should the agent cast?; (2) how long should the agent spend casting before surging upwind?; (3) where should the agent sniff during the casting phase? Specifically, we highlight and quantify the various trade-offs associated with the cast-sniff-surge modes of exploration.

To introduce the main simplification of the model, we note that in the exploratory regime at large distances, the agent is more likely to detect odor by sniffing in the air due to the larger detection range of airborne cues. We therefore ignore odor signal on the ground and assume the agent only detects odor by sniffing in the air. This simplifies the analysis considerably as the search path is then parameterized by the discrete locations at which the agent sniffs in the air rather than the specific trajectory taken between sampling locations. The agent's prior belief distribution, $\mathbf{b}(x, y)$, of its location with respect to the source is assumed to be uniform with length $L_x$ ($\gg x_{\text{thr}}$) (along the downwind direction) and width $L_y$ ($\gg y_{\text{thr}}$), similar to the example shown in *Figure 5* (top). The probability of detecting an odor signal in a sniff, $r(x, y)$, depends on the range and width of the plume through the parameters $x_{\text{thr}}$ and $y_{\text{thr}}$ respectively, which we assume are known to the agent. To decouple the upwind surge and crosswind cast, we approximate the detection probability map in *Figure 2(c)* as $r(x, y) = f(x)g(y)$, where $f(x)$ is a constant when $0 < x < x_{\text{thr}}$ and 0 otherwise and $g(y)$ has a characteristic length scale $y_{\text{thr}}$, namely, $g(y) = e^{-y^2/2y_{\text{thr}}^2}$. Instead of receiving reward for finding the source, the agent receives a unit reward (discounted at a rate $\lambda$) on detecting odor, at which point the search ends.

We first build an intuitive picture of the search dynamics in this simplified setting. To detect the plume, the agent has to sufficiently explore the region in which it expects to detect odor, which is delineated by the prior. Each sniff in the air will sample a patch of size $\sim x_{\text{thr}} \times y_{\text{thr}}$ immediately upwind from its location. Since the prior's width is larger than the plume width ($L_y \gg y_{\text{thr}}$), the agent cannot sample the full breadth of the prior in one sniff and will thus have to cast crosswind and sniff the air. As each sniff explores a patch of length $x_{\text{thr}}$, a few bouts of cast-and-sniff across a width $L_y$ will effectively explore an upwind region of size $x_{\text{thr}} \times L_y$. If odor is not detected during the bouts, the likelihood that the source is contained within this region is negligible. An application of Bayes' rule converts the initial prior of length $L_x$ into a posterior of reduced length $L_x - x_{\text{thr}}$. Since the agent now believes that it is downwind of the plume by at least a distance $x_{\text{thr}}$, it will surge upwind by $x_{\text{thr}}$ and explore the next patch using the same procedure of cast-and-sniff. The process is repeated until the plume is detected.

The above argument implies that the search process can be split into distinct episodes where in each episode the agent executes bouts of cast-and-sniff followed by a surge of length $x_{\text{thr}}$. The search can then be decomposed into a maximum of $N \sim L_x/x_{\text{thr}}$ distinct episodes or until the agent detects odor. Suppose that in each episode $n$ ($n = 1, 2, \ldots, N$), the agent spends time $t_n$ executing bouts of cast-and-sniff. The casting duration $t_n$ is to be optimized. As discussed next, the optimal $t_n$ depends on

the cumulative probability of not detecting the signal (conditional on the target being in that patch) after casting for time $t$, denoted $c(t)$, which in turn depends on the sampling strategy during casting.

The expected discounted reward at the beginning of the search is $V_1 \equiv \langle e^{-\lambda T} \rangle_T$, where $T$ is the time taken to find the odor signal. We use dynamic programming to compute and optimize $V_1$. $V_1$ is the sum of the expected reward if the agent finds odor in the first patch within time $t_1$ and the expected reward after moving to the next patch if it does not. The information gained from not detecting odor is taken into account in the latter term through a Bayesian update of the prior. However, we show that $V_1$ and the casting times, $t_1, t_2, \ldots, t_N$, can be calculated using an equivalent, simpler expression which does not require Bayesian updates (Materials and methods). Denote $V_n$ as the expected discounted reward at the beginning of the $n$th episode. We show that $V_1$ can be calculated using the recursive equation

$$V_n = \max_t \left\{ -\frac{1}{N} \int_0^t c'(s)e^{-\lambda s}ds + e^{-\lambda\left(t+\frac{x_{\text{thr}}}{v}\right)}V_{n+1} \right\}.$$

(2)

The time $t$ that maximizes the parenthesis determines the optimal duration $t_n$ the agent should spend casting before surging upwind. The first and second terms in the parenthesis of (2) are the expected discounted rewards if the agent detects odor during casting and the search ends or if it does not detect odor, surges a distance $x_{\text{thr}}$ (which takes time $x_{\text{thr}}/v$) and continues to the next episode, respectively. The factor $-c'$ in the first term is the probability density to detect odor at time $t$ conditional on the target being in the current patch, which depends on the casting strategy discussed further below. Since the prior is uniform, the probability that the target is in the current patch is $1/N$, which sets the factor in front of the integral.

We first show that the duration $t_n$ obeys a marginality condition. The agent should stop casting when the value of continuing to explore the current patch is just outweighed by the value of moving on to the next patch. This intuition is quantified by the optimization over $t$ in *Equation 2*. Zeroing the time derivative of *Equation 2*, we obtain that $t_n$ is the value of $t$ that satisfies the equality $-c'(t)e^{-\lambda t}/N = \lambda e^{-\lambda\left(t+\frac{x_{\text{thr}}}{v}\right)}V_{n+1}$. The left-hand side is the rate of value acquired for staying in the current patch. The right-hand side is the rate of value lost for delaying departure. Thus, by maximizing value we obtain that, at optimality, the added value of continuing to cast matches the added value of anticipating surge, similar to the condition prescribed for patch-leaving decisions during foraging by marginal value theory (MVT) (*Charnov, 1976*). The marginality condition leads to a relationship between the casting time and the value at the next episode

$$-c'(t_n) = N\lambda e^{-\lambda x_{\text{thr}}/v}V_{n+1}.$$

(3)

When $n = N$, the agent casts indefinitely, which gives $V_N = -\frac{1}{N}\int_0^\infty c'(s)e^{-\lambda s}ds$ from *Equation 2*. The casting time for each episode is obtained using the boundary condition *Equation 3* and $c(t)$, which we shall determine in the next paragraph. Note that we have ignored the possibility that after the $N$th episode, the agent turns back and moves downwind to re-explore earlier regions, which can be incorporated into this framework by imposing a different boundary condition. This extension marginally affects the earlier stages of the search path and does not affect general conclusions.

We now optimize for the sampling strategy during the bouts of cast-and-sniff, which in turn determines $c(t)$. Each cast-and-sniff requires the agent to decide where to sniff on the crosswind axis, given the marginal posterior distribution $\tilde{\mathbf{b}}(y) = \int_0^{x_{\text{thr}}} dx\, \mathbf{b}(x, y)$. The next sniff location at a displacement $\Delta y$ from the current location is obtained from a dynamic programming equation similar to *Equation 2*, which relates the current value to the value of moving and sampling elsewhere.

$$V(\tilde{\mathbf{b}}) = \max_{\Delta y} \left\{ \left[ \Gamma_{\tilde{\mathbf{b}}}(\Delta y) + (1 - \Gamma_{\tilde{\mathbf{b}}}(\Delta y))V(\tilde{\mathbf{b}}') \right] \times e^{-\lambda\left(|\Delta y|/v + t_{\text{sniff}}\right)} \right\},$$

(4)

where $\tilde{\mathbf{b}}'$ is the posterior after sampling at the new location conditional on no detection, and $\Gamma_{\tilde{\mathbf{b}}}(\Delta y)$ is the probability of detection. The two terms in the Bellman equation correspond to the cases when the agent detects odor and does not detect odor, respectively, which are discounted in proportion to the time taken to travel a distance $|\Delta y|$, $|\Delta y|/v$, and the time taken to sniff in the air, denoted by $t_{\text{sniff}}$. At each decision, the recursion in *Equation 4* is expanded and optimized with respect to the subsequent $n_{\text{steps}}$ sniff locations $\Delta y_1, \Delta y_2, \ldots, \Delta y_{n_{\text{steps}}}$ using standard gradient-free optimization methods.

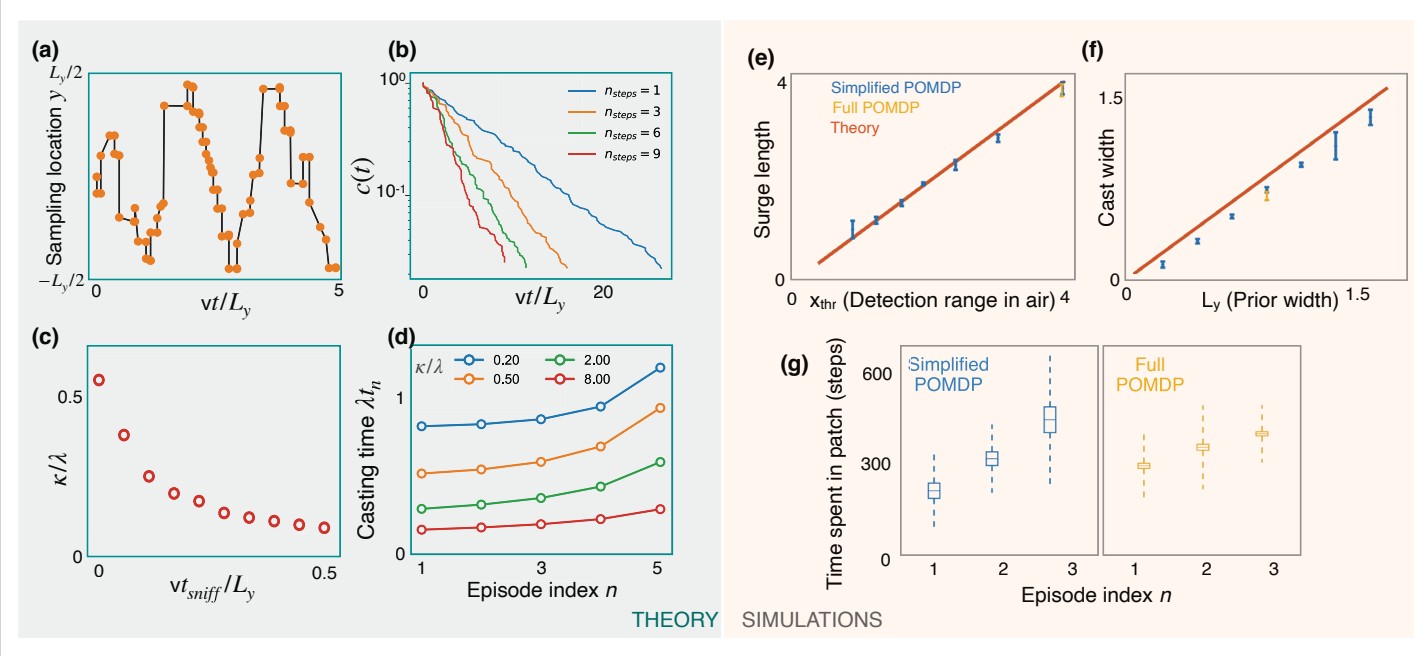

**Figure 6.** Predictions of a simplified POMDP with detections allowed in the air only. Results are both shown for an analytical (green, left) and a computational (orange, right) model. (a) Optimized sniff locations during casting (conditional on no detection) show a zigzag of increasing amplitude. (b) The probability of not detecting the signal against time, $c(t)$, decays exponentially with detection rate, $\kappa$, shown here for different values of the optimization depth $n_{steps}$. $\kappa$ saturates beyond $n_{steps} = 9$. (c) $\kappa$ monotonically decreases with the time per sniff, $t_{sniff}$, reflecting the cost of pausing to sniff the air. In panels (a), (b), and (c), we use $y_{thr}/L_y = 1/20$, $L_y = 1$, $\lambda = 0.5$, $v = 1$, and $t_{sniff} = 0$ (for (a) and (b)). (d) Casting times (in units of $1/\lambda$) generally increase as the search progresses. Obtained using **Equation 5** for different values of $\kappa/\lambda$ (colored lines). Here, $N = 6$ and $\lambda x_{thr}/v = 0.05$. (e,f) The surge length and cast width from simulations of a simplified partially observable Markov decision process (POMDP), where the agent can detect an odor signal only by sniffing in the air. Results for different prior and plume dimensions (blue stars) align with the theoretical prediction (red line) that the surge length and cast width are equal to the detection range in air, $x_{thr}$, and the prior width, $L_y$, respectively. Results from the full POMDP, where the agent can detect odor on the ground, are also consistent with the predictions (yellow crosses). Experiments were repeated over 5 different seeds. (g) The time spent casting in each patch for the simplified and full POMDP increases as the search progresses, as predicted by the theory (panel (d)). Here, we set the prior length, $L_x = 4x_{thr}$, which corresponds to $N = 4$ patches. Boxes and dashed lines represent the standard error and the standard deviation around the mean, respectively.

The agent then moves by $\Delta y_1$, $\tilde{\mathbf{b}}$ is updated using Bayes' rule and the procedure is repeated. Optimization yields a sampling strategy and the corresponding $c(t)$.

The optimized casting strategy is a zigzag (**Figure 6a**) which expands over time to the width of the prior. The expanding casting width can be again interpreted in the marginal value framework: at the first counterturn, the value of finding odor toward the agent's previous heading is just outweighed by the value of finding it on the opposite side after appropriate discounting due to travel time. The discounted value on the opposite side can match the value at the current side due to the imbalance in the probabilities of detection between the two sides after the Bayesian update. This idea extends to subsequent counterturns, until the zigzags extend to the prior's width. The probability of not detecting odor decays exponentially with a rate that increases with optimization depth $n_{steps}$ (**Figure 6b**), highlighting the importance of a long-term strategy. In the low-detection rate limit, we generically expect a constant detection rate (say $c(t) = e^{-\kappa t}$), consistent with the exponential decay observed in simulations. The detection rate $\kappa$ decreases with $t_{sniff}$ (**Figure 6c**), which in turn translates to a decreased value (from **Equation 3**) and highlights the cost of pausing to sniff in the air. From **Equation 3**, we then have

$$t_n = \kappa^{-1} \log\left(\frac{\kappa e^{\lambda x_{thr}/v}}{N\lambda \, V_{n+1}}\right). \tag{5}$$

We use **Equations 2 and 5** along with the boundary condition $V_N = \frac{\kappa}{N(\kappa+\lambda)}$ to solve for the casting times. The results show increasing casting times with episode index $n$ (**Figure 6d**). Intuitively, as the search progresses, the marginal cost for the agent to continue casting decreases due to its increasing

confidence that it is in the right patch, driving the agent to spend more time casting before leaving the patch.

We test predictions from the theory using simulations of a simplified POMDP. Specifically, the agent is trained to find the target with an odor signal that can be detected only by sniffing in the air. The detection probability map $r(x, y)$ is rectangular with plume detection range $x_{thr}$ and width $y_{thr}$. Simulations confirm that the surge length and cast width are equal to the detection range and the prior width, respectively (*Figure 6e and f*). The time spent exploring a patch increases monotonically as the search progresses, as predicted by the theory (*Figure 6d and g*). Notably, we find that the trajectories from the full POMDP considered in the previous sections are also consistent with these predictions, suggesting that these aspects are generic features of foraging for a sparse odor signal during the first phase of exploration.

## Discussion

Motivated by the goal of disentangling elementary components in the complexity of animal behavior, we have investigated a dynamics driven entirely by olfactory cues. In the model, an agent searches for a source of odors transported by a turbulent flow and at each step decides either to move while sniffing on the ground or pause, rear up and sniff in the air. The goal is to locate the source in the shortest possible time, which is the reward function used to identify effective policies of action using machine learning methods. Analogously to dogs and rodents mentioned in the introduction, we obtain behavioral policies which feature alternation between the two modalities of sniffing on the ground vs. in the air. The appeal of our approach is that we could identify the rationale for the observed alternation and its basic factors. On the one hand, movement and progression toward the source is halted during the rearing phase of sniffing in the air. On the other hand, odor sources create large turbulent plumes that reach larger distances in the air than on the ground. Therefore, sniffing in the air may have a higher chance of intersecting odor cues than on the ground. These two competing effects underlie the process of alternation and their balance determines the rate of switching between the two modalities, which depends on the distance as discussed in the next paragraph.

The effect of alternation is particularly pronounced at large distances to the source. There, due to turbulent mixing, the odor concentration drops substantially and no gradients are present (*Celani et al., 2014*). In our realistic setting, where the searcher does start at large distances, the process can be qualitatively split in two phases: first, the agent needs to approach the source enough for an almost continuous odor plume to be present; second, it needs to locate the source within the plume. The latter task, which is the regime that most laboratory experiments have considered so far (*Reddy et al., 2022*), is much easier than the former as the rate of odor detection close to the source and within the conical plume is relatively high. Therefore, the task boils down to staying close to the center of the conical plume, where the signal is highest. Conversely, the bottleneck during the first, harder phase is the scarcity of information on the location of the source, which the agent tries to overcome by increasing its chances of odor detection. Slowing down its progression is thus the price that the agent pays in order to get oriented in the uncertain conditions typical of large distances to the source. The transition between the two search phases typically occurs after a handful of odor detections.

Note that we have focused here on the case of a stationary source, where odor statistics in the air and on the bottom layers are discriminated by the adsorption on the ground. In fact, at the onset of odor emission (and even in the absence of adsorption), plumes start out larger in the air than near the ground, simply because air travels more slowly near the ground. It follows from our results that alternation should be more frequent in the early stages of odor release in non-steady conditions. This prediction could be tested experimentally by switching on an odor source and monitoring the fraction of sniffing in the air as a function of the time elapsed since the switch and the onset of odor emission. We expect that the benefits of sniffing the air vs. the ground will hold even if the adsorption of odor molecules is modified. While the fully absorbing conditions considered here clearly reduce the amount of odors close to the ground, the odors will extend less on the ground than in the air even for other boundary conditions where partial absorption is considered. We expect then that quantitative properties of the odor will depend on details of the adsorption process but alternation and its increase with the distance to the source will qualitatively hold in general.

While we considered here two olfactory sensorimotor modalities, we expect that our methodology and results apply more broadly to distinct sensory systems and cues. If there is no conflict in the

acquisition and processing of multiple sensory cues, then it is clearly advantageous to combine them. Conversely, if their combination has some form of cost and a partial or total conflict exists, then our results suggest that the logic identified here will apply and there will be some form of alternation.

The machine learning methodology that we have employed here to identify effective policies of actions belongs to the general family of POMDP (*Kaelbling et al., 1998*; *Sutton and Barto, 2018*). This framework applies to a broad class of decision problems, where agents need to accomplish a prescribed task by a series of actions taken with partial knowledge of the environment. Specifically, the agent combines external cues and its internal model of the world to infer a belief about the state of the environment. In our setting, the agent is the searcher, cues are odor detections (or their absence), the task is to localize the source, and beliefs pertain to the location of the source of odors. While the agent proceeds along its path and gathers information via odor cues, its belief narrows down and eventually concentrates at the location of the source. Trajectories of a POMDP agent with a single sensory modality and their relation to phenomenological approaches as *Vergassola et al., 2007*, were discussed in *Reddy et al., 2022*. Here, we have given the agent the choice of multiple sensory modalities at each decision step, which allowed us to highlight the presence of alternation and establish its link with MVT (*Charnov, 1976*).

MVT describes the behavior of an optimally foraging individual in a system with spatially separated resources. Due to the spatial separation, animals must spend time traveling between patches. Since organisms face diminishing returns, there is a moment the animal exhausts the patch and ought to leave. In MVT, the optimal departure time is determined as the time at which the marginal value of staying in a patch equals that of leaving and exploring another patch. In our setting, these patches correspond to regions of the agent's belief which are explored using a combination of casting and sniffing in the air. MVT thus determines when to stop cast-and-sniff exploration and surge toward the next patch in the belief.

As MVT, our machine learning methodology provides a normative theory that disregards constraints by construction, for instance on memory of the agent or its capacity of processing data. The goals of the approach are threefold: first, it provides understanding and fundamental limits on performances; second, understanding can inspire the development of strategies less demanding, as it was the case with *Vergassola et al., 2007*, and *Masson, 2014*; third, the developed methodologies apply to robots, which are typically less constrained than animals (*Russell, 1999*; *Webb and Consilvio, 2001*). The counterpart of cognitive strategies like those developed here is provided by reactive strategies, which take the opposite view of enforcing strong constraints, for example, on memory, and explore what can be achieved within those limits. The two approaches provide complementary views to the problem and we refer to the work in *Voges et al., 2014*, for a discussion of their comparison, values, and limits. While a similar study for the case of alternation is beyond the scope of this work, a few comments on memory requirements and robustness of our strategies are worth. The increase of the number of hyperplanes with the number of training episodes is shown in *Figure 3—figure supplement 1*. As in the case of olfactory searches mentioned above, we expect that simplified strategies with less memory requirements will be inspired by our algorithms. The robustness of our strategies is investigated in *Figure 4—figure supplement 2*. The plots show the performance of our search strategy when the model of the environment is incorrect. Robustness with respect to the length and steadiness of the model are reported. Specifically, training is performed with a model that is consistent with the stationary plume used in the main text. The agent is then tested for localization of the source of a larger or smaller plume (*Figure 4—figure supplement 2* left, for different sizes of the plume) or of a meandering plume (i.e. identical in size to the plume used for training, but rotating with different frequencies and amplitudes, *Figure 4—figure supplement 2* right). Note that while the plume varies in these simulations, the agent still keeps the static model used for training, thus the model is now inaccurate. The upshot is that strategies developed here do not need to have their parameters set exactly at the correct values of the environment and are robust to partial misrepresentations.

We conclude by noting that, in addition to the familiar cases of dogs and rodents mentioned in the introduction, other species can sense chemical cues both in the bulk and on surfaces, and may feature a similar phenomenology of alternation. In particular, a large body of experimental evidence has been collected for turbulent plume-tracking by aquatic organisms, as reviewed in *Webster and Weissburg, 2009*. Crustaceans sense chemical cues with their antennules floating in water and switch to sensing with their feet as they approach the target (*Grasso, 2001*). For example, lobsters were

observed in dim light in a flume of dimensions 2.5 m × 90 cm × 20 cm, as they left their shelter upon release of a turbulent plume of odor obtained from grounded mussel (*Moore et al., 1991*). As the animals encountered the plume, they often displayed special behaviors, including raising up, sweeping their sensory legs on the bottom of the flume, and increasing flicking of lateral antennules. Similar observations were made for blue crabs capturing live clams or tracking spouts releasing clam extract (*Weissburg and Zimmer-Faust, 2002*). In these experiments blue crabs would occasionally lower their abdomen closer to the surface or extend their walking legs to raise above their normal height. Finally, pelagic marine mollusks *Nautilus pompilius,* were observed to track the source of a turbulent plume by swimming at different heights, above and below the center of the plume. Interestingly, most animals sampled at higher heights beyond 1 m from the source, and swam at lower heights when closer to the source (*Basil and Atema, 2000*). These experiments indicate that animals may alternate between different heights, and that sampling at higher elevation may be particularly useful at larger distances, which is again in qualitative agreement with our results. The ensemble of these observations suggest that alternation between sensorimotor modalities is likely to be present in the behavior of aquatic organisms as well. We hope that results presented here will motivate more experiments, on dogs, rodents, and aquatic organisms alike, with the goal of assessing quantitative aspects of the observed behaviors, testing our framework and advancing understanding of how sensorimotor modalities are integrated.

## Materials and methods

### Direct numerical simulations

The Navier-Stokes (*Equation M1*) and the advection-diffusion equation for passive odor transport (*Equation M2*) describe the spatiotemporal evolution of odor released in a fluid. We can solve these equations with DNS and obtain realistic odor fields to feed the POMDP algorithm:

$$\partial_t \mathbf{u} + \mathbf{u} \cdot \nabla \mathbf{u} = -\tfrac{1}{\rho} \nabla P + \nu \nabla^2 \mathbf{u} \qquad \nabla \cdot \mathbf{u} = 0 \tag{M1}$$

$$\partial_t \theta + \mathbf{u} \cdot \nabla \theta = \kappa_\theta \nabla^2 \theta + q, \tag{M2}$$

where $\mathbf{u}$ is the velocity field, $\rho$ is the fluid density, $P$ is pressure, $\nu$ is the fluid kinematic viscosity, $\theta$ is the odor concentration, $\kappa_\theta$ is its diffusivity, and $q$ an odor source.

We simulate a turbulent channel of length $L$, width $W$, and height $H$, where fluid flows from left to right and hits a solid hemicylindrical obstacle of height 38 cm set on the ground, which produces turbulence. A horizontal parabolic velocity profile is set at the left boundary $u = 6U_b \left[ \frac{z}{H} - \left( \frac{z}{H} \right)^2 \right]$,

**Table 1.** Parameters of the simulation.
Length $L$, width $W$, height $H$ of the computational domain; horizontal speed along the centerline $U$; mean horizontal speed $U_b = \langle u \rangle$; Kolmogorov length scale $\eta = (\nu^3/\epsilon)^{1/4}$, where $\nu$ is the kinematic viscosity and $\epsilon$ is the energy dissipation rate; mean size of grid cell $\Delta x$; Kolmogorov timescale $\tau_\eta = \eta^2/\nu$; energy dissipation rate $\epsilon = \nu/2 \langle (\partial u_i/\partial x_j + \partial u_j/\partial x_i)^2 \rangle$; Taylor microscale $\lambda = \sqrt{\langle u^2 \rangle / \langle (\partial u/\partial x)^2 \rangle}$; wall length scale $y^+ = \nu/u_\tau$, where the friction velocity is $u_\tau = \sqrt{\tau/\rho}$ and the wall stress is $\tau = \rho \nu du/dz|_{z=0}$; Reynolds number $Re = U(H/2)/\nu$ based on the centerline speed $U$ and half height; Reynolds number $Re_\lambda = U\lambda/\nu$ based on the centerline speed and the Taylor microscale $\lambda$; magnitude of velocity fluctuations $u'$ relative to the centerline speed; large eddy turnover time $T = H/2u'$. First and third rows are the labels, second and forth rows report results in non-dimensional units, third and fifth rows correspond to dimensional parameters in air, assuming the mean speed is 25 cm/s.

| $L$ | $W$ | $H$ | $U$ | $U_b$ | $\eta$ | $\Delta x$ | |
|---|---|---|---|---|---|---|---|
| 40 | 8 | 4 | 32 | 25 | 0.006 | 0.025 | |
| 15 m | 3 m | 1.5 m | 0.33 m/s | 0.25 m/s | 0.23 cm | 1 cm | |
| $\tau_\eta$ | $\epsilon$ | $\lambda$ | $y^+$ | $Re$ | $Re\lambda$ | $u'/U$ | $T$ |
| 0.01 | 39 | 0.17 | 0.004 | 16000 | 1370 | 10% | $64\tau_\eta$ |
| 0.36 s | 1.2e-4 m²/s³ | 6 cm | 0.14 cm | | | | |

where $z$ is the vertical coordinate and $U_b$ is the mean horizontal speed. We impose the no-slip condition at the ground and on the obstacle and an outflow condition at the other boundaries (see *Rigolli et al., 2021*, for more details).

When the turbulent flow is fully developed a concentrated odor source is added at 0.58 m from the ground, that is, 20 cm above the center of the obstacle. The source is defined by a Gaussian profile with radius $\sigma \sim 5\eta$, where $\eta$ is the smallest scale of turbulent eddies (see *Table 1*). We set adsorbing boundary conditions at the inlet, on the ground, and on the obstacle and zero gradient conditions on the sides and top.

The simulation was realized by customizing the open-source software Nek5000 (*Fischer et al., 2008*) developed at Argonne National Laboratory, Illinois. The three-dimensional (3D) volume of the channel is discretized in a finite number of elements and Nek5000 solves the Navier-Stokes and scalar transport equations within every element with a spectral element method. To accurately describe all relevant scales of turbulence from the dissipative scale to the length of the domain, the solution is expanded in eighth grade polynomials in each of 160,000 elements, thus effectively discretizing space in 81,920,000 grid points. *Table 1* summarizes the parameters that characterize the flow. Each DNS runs for 300,000 time steps where $\delta t = 10^{-2}\tau_\eta$ following a strict Courant criterionwith $U\Delta t/\Delta x < 0.4$ to ensure convergence of both the velocity and scalar fields. Snapshots of velocity and odor fields are saved at constant frequency $\omega = 1/\tau_\eta$. Fully parallelized simulations require 2 weeks of computational time using 320 cpu, see *Rigolli et al., 2021*, for further details. The dataset containing odor concentration fields at nose and ground level is publicly available at https://zenodo.org/record/6538177#.Yqrl_5BByJE.

## The POMDP framework

We briefly introduce POMDPs before describing the specific algorithms used in our simulations. We refer to *Shani et al., 2013*, for a detailed review on POMDPs. POMDPs are a generalization of Markov decision processes (MDPs) analogous to the relationship between hidden Markov models and Markov models (*Kaelbling et al., 1998*; *Sutton and Barto, 2018*). In an MDP, we define a state space, an action space, and a reward function. The dynamics of the state space is Markovian and is defined entirely by the transition matrix, $T(s'|s, a)$, which gives the probability of transitioning to state $s'$ given the current state $s$ and the action taken, $a$. After each transition, the agent receives a reward, which has expectation $r(s, a, s')$. Given the transition matrix and the reward function, the goal is typically to find the unique optimal policy, $\Pi^*(s)$, which maximizes the discounted sum of future rewards, $\langle r_0 + \gamma r_1 + \gamma^2 r_2 + \ldots \rangle$, where $\gamma$ is the discount factor and $r_t$ is the expected reward $t$ steps after the initial state. Often, but not always, this involves solving for the value function, $V(s)$, which is the expected discounted sum of rewards from state $s$, conditional on policy $\Pi^*(s)$. The value function satisfies the central dynamic programming equation known as the *Bellman, 2003*, equation:

$$V(s) = \max_a \left( \sum_{s'} T(s'|a, s) \left\{ r(s, a, s') + \gamma V(s') \right\} \right). \tag{M3}$$

While MDPs deal with fully observable states, POMDPs have one additional feature which makes it appropriate for our setting. Instead of observing the current state, the agent only receives certain observations, $o$, from which the true latent state has to be dynamically inferred. The agent is assumed to have a model of the environment, $P(o|s, a)$. In our setting, this likelihood function encodes the statistics of detections and non-detections at various locations downwind of an odor source (*Figure 2—figure supplement 1*). A POMDP therefore maps a sequence of recent observations and actions $o_{-1}, a_{-1}, o_{-2}, a_{-2}, \ldots$ to a strategic action. While the dimensionality increases rapidly with the length of the observation history, the entire history is encoded by the current posterior distribution over states, $\mathbf{b}$, also known as the *belief vector*. The problem of solving for the optimal action is recast as the problem of solving for the policy $\Pi^*(\mathbf{b})$. The Bellman equation on states for MDPs translates into a Bellman equation on belief vectors for POMDPs:

$$V(\mathbf{b}) = \max_a \left( \sum_{s,s'} \mathbf{b}(s)T(s'|a, s) \right.$$
$$\left. \times \left\{ r(s, a, s') + \gamma \sum_o P(o|s', a)V(\mathbf{b}^{a,o}) \right\} \right), \tag{M4}$$

where $\mathbf{b}^{a,o}$ is the posterior belief state given the agent takes action $a$ and observes $o$. Using Bayes' rule, $\mathbf{b}^{a,o}$ is given by

$$\mathbf{b}^{a,o}(s') = P(s'|a, o, \mathbf{b}) = \frac{P(o|s',a)\sum_s T(s'|s,a)\mathbf{b}(s)}{P(o|\mathbf{b},a)}, \tag{M5}$$

where the normalizing factor is

$$P(o|\mathbf{b}, a) = \sum_{s'} \mathbf{b}(s') \sum_s T(s|s', a)P(o|s, a). \tag{M6}$$

Recall that $\mathbf{b}(s) = P(s)$ and thus the term $\sum_s T(s'|s, a)\mathbf{b}(s)$ is the probability of transitioning to state $s'$ given the current belief state $\mathbf{b}$ and action $a$ whereas $P(o|s', a)$ is the probability of observing $o$ at the new state $s'$. Intuitively, the Bayes' rule takes into account the new information gained from the most recent observation (via $P(o|s', a)$) and the information lost due to the state space dynamics (via $T(s'|s, a)$), which are in turn influenced by the action.

## Algorithms to solve POMDPs

We use POMDP-solvers which approximate the value function, $V(\mathbf{b})$, for all $\mathbf{b}$. If the value function is known, the optimal policy is simply to choose the action that yields the highest future expected return given the current belief vector:

$$\Pi^*(\mathbf{b}) = \arg\max_a \quad \sum_{s,s'} \mathbf{b}(s)T(s'|a, s)$$
$$\times \left\{ r(s, a, s') + \gamma \sum_o P(o|s', a)V(\mathbf{b}^{a,o}) \right\}. \tag{M7}$$

Computing the value function $V(\mathbf{b})$ exactly for all belief vectors for tasks containing more than a handful of states is infeasible. Existing methods exploit a specific representation of the value function, which leads to the approximation discussed by *Shani et al., 2013*. We recapitulate here the main results and refer to *Shani et al., 2013*, for more details. In particular, it can be shown that the value function can be approximated arbitrarily well by a finite set $H$ of hyperplanes (*Sondik, 1978*), each of which is parameterized by $\alpha(s)$:

$$V(\mathbf{b}) = \max_{\alpha \in H} \alpha \cdot \mathbf{b}. \tag{M8}$$

An initial set $H$ is expanded using the Bellman *Equation M4*. Using vector notation, we can write

$$V(\mathbf{b}) = \max_a \left\{ \mathbf{r}_a \cdot \mathbf{b} + \gamma \sum_o P(o|\mathbf{b}, a)V(\mathbf{b}^{a,o}) \right\}, \tag{M9}$$

where $\mathbf{r}_a(s) \equiv \sum_{s'} T(s'|a, s)r(s, a, s')$. Let $\alpha^{a,o}(s)$ be defined as

$$\alpha^{a,o}(s) = \sum_{s'} \alpha(s')P(o|s', a)T(s'|s, a). \tag{M10}$$

and using the belief update (*Equation M5*), it follows that

$$V(\mathbf{b}) = \max_a \left\{ \mathbf{r}_a \cdot \mathbf{b} + \gamma \sum_o \max_\alpha \mathbf{b} \cdot \alpha^{a,o} \right\}. \tag{M11}$$

Given the previous set $H$, we can add a new $\alpha$ vector to it corresponding to belief vector $\mathbf{b}$ called the 'backup' operation:

$$\text{backup}(H, \mathbf{b}) = \arg\max_{\alpha_a^\mathbf{b}} \mathbf{b} \cdot \alpha_a^\mathbf{b}, \tag{M12}$$

$$\text{where} \quad \alpha_a^\mathbf{b} = \mathbf{r}_a + \gamma \sum_o \arg\max_{\alpha^{a,o}} \mathbf{b} \cdot \alpha^{a,o}. \tag{M13}$$

In other words, given a previous set $H$ and new belief vector, one can use the Bellman equation to update $H$ and obtain a better approximation to the value function. The key computational advantage of using the above backup operation is that the $\alpha^{a,o}$'s can be pre-computed for the current $H$ and re-used when backing up.

The question then is: how do we efficiently collect new belief vectors to update $H$ and prune vectors from $H$ that are no longer necessary? Algorithms differ at these two stages. We use *Spaan and Vlassis, 2005*, which simulates random exploration of the agent. Specifically, at each step in a 'training' episode, we start from an initial prior, pick actions (uniform) randomly and then sample

observations from $P(o|\mathbf{b}, a)$. The new belief vector obtained using Bayes' rule is then used to backup $H$. Finally, after adding a new set of $\alpha$ vectors into $H$, it is efficient to prune the existing ones that are guaranteed to not be used. We prune the $\alpha$ vectors whose every component is smaller than those of another vector (see *Shani et al., 2013*, for other heuristic pruning methods).

Three parameters can be tuned: the discount factor $\gamma$, the number of belief points sampled per each episode of random exploration, and the total number of training episodes. The discount rate sets the planning horizon, which is set to be of the same order as the typical number of steps to get to the target. Increasing the latter two parameters improves the strategy at the expense of increased training time. We solve the POMDP for various values of these two parameters and show that the performance saturates at a parameter range within computational feasibility.

## POMDPs for learning sniff-and-search strategies

To implement POMDPs that learn to navigate odor plumes by employing multiple modes of search, we consider a simple state space consisting of a 2D grid with dimensions $10 \times 2$ discretized with 30 points per unit length so that the state space has size $10 \times 2 \times 30^2 = 18,000$. The agent can take six possible actions corresponding to movement in either of the four directions or staying at the same spot while sniffing ground odor cues. The sixth action corresponds to staying at the same spot and sniffing airborne odor cues. After every action, the agent can make one of three observations – no detection, an odor detection, or finding the odor source. Odor detections are binarized, that is, the odor is detected if the concentration is above a certain threshold. The intermittent nature of turbulent fluctuations imply that there is little additional information in the graded concentration beyond the information contained in the detection rate (*Vergassola et al., 2007*; *Victor et al., 2019*). We assume a Poisson rate of detection with the rate map at the ground level and at the nose level when sniffing in the air obtained by measuring the fraction of time the odor concentration is above 0.14% with respect to the maximum concentration at source in the flow simulations. As described above, we use the Perseus algorithm (*Spaan and Vlassis, 2005*), which performs random exploration starting from a given prior belief of where the source is located. We use a uniform prior of dimensions $28.6\,\mathrm{m} \times 3.4\,\mathrm{m}$. After training, the POMDP algorithm yields a set $H$ which encodes an approximation to the value function mapping belief vectors to expected discounted rewards for each of the possible actions. The decision at each step is then obtained from (*Equation M7*).

## Parameters for POMDP used in main text

The main figures represent results using: discount factor $\gamma = 0.99$, number of training episodes $i = 320$, number of belief points sampled per training episode $i' = 100$, likelihood in the air and at the ground is defined as shown in *Figure 2—figure supplement 1*, in *Figure 6* likelihood in the air is defined as a rectangle with dimensions $x_{\mathrm{thr}} \times y_{\mathrm{thr}}$. Results in *Figure 4* are tested and averaged over three different starting positions (x=8; y=0, 0.3, –0.5), 8 different seeds, 50 different realizations for the same seed (trajectories differ for the history of detections according to the Poissonian model).

The effect of varying $\gamma$ are represented in *Figure 4—figure supplement 1*. (all other parameters are kept constant); the effect of varying the number of training episodes is represented in *Figure 3—figure supplement 1* (averaged over three different locations and three seeds): few tens of training episodes are enough to successfully locate the target, to obtain a robust strategy and to avoid the poor performance showed in the first two violin plots of *Figure 3—figure supplement 1B*, it is necessary to increase the number of training episodes. *Figure 3—figure supplement 1C* shows the cost of increasing the number of episodes in terms of memory (i.e. the number of $\alpha$ vectors the agent has to store during the training phase).

Training requires up to 2 days in time on one processor, while testing a single realization takes $\sim 10$ hr.

## Derivation of (*Equation 2*)

We consider a scenario where a target is located at one of $N$ possible patches, $n = 1, 2, \ldots, N$ with probabilities $\mathbf{p}_0 = (p_1, p_2, \ldots, p_N)$. Note that $p_n = 1/N$ for all $n$ for the prior considered in the main text. The agent starts at $n = 1$ and moves sequentially from $n = 1$ to $n = N$ while spending time $t_n$ sampling in each patch. Moving from a patch to the next one takes time $\tau \equiv x_{\mathrm{thr}}/v$. At $n = N$, the agent samples indefinitely, $t_N = \infty$. The agent receives reward of one when the target is found in a patch, which is

discounted at rate $\lambda$. The value $V_1 \equiv \langle e^{-\lambda T} \rangle_T$, where $T$ is the search time, is the expected discounted reward optimized w.r.t. $t_n$'s. We derive two sets of recursive equations (with and without Bayesian updates) to calculate $V_1$. We show that both formulations lead to the same optimal casting times, however, the set of equations without Bayesian updates are much simpler to compute.

Suppose the cumulative probability of finding the target in time $t$ *conditional* on the target being in that patch is $d(t)$. Note that $c(t) \equiv 1 - d(t)$ is used in the main text. Denote $r(t) \equiv \int_0^t d'(s)e^{-\lambda s}ds$. This is the expected discounted reward if the agent searches for time $t$ in a patch that contains the target.

Say $N = 3$. Since $V_1$ is the expected discounted reward optimized over the casting times $t_1, t_2$, we have

$$
\begin{aligned}
V_1 &= \max_{t_1,t_2} \left\{ p_1 r(t_1) + e^{-\lambda(t_1+\tau)} p_2 r(t_2) + e^{-\lambda(t_1+t_2+2\tau)} p_3 r(\infty) \right\} \\
&= \max_{t_1,t_2} \left\{ p_1 r(t_1) + e^{-\lambda(t_1+\tau)} \left( p_2 r(t_2) + e^{-\lambda(t_2+\tau)} \left( p_3 r(\infty) \right) \right) \right\} \\
&= \max_{t_1} \left\{ p_1 r(t_1) + e^{-\lambda(t_1+\tau)} \right. \\
&\qquad \left. \times \max_{t_2} \left\{ p_2 r(t_2) + e^{-\lambda(t_2+\tau)} \left( p_3 r(\infty) \right) \right\} \right\}
\end{aligned}
$$
(M14)

The last equation above motivates a recursive equation for general $N$:

$$
V_n = \max_{t_n} \left\{ p_n r(t_n) + e^{-\lambda(t_n+\tau)} V_{n+1} \right\},
$$
(M15)

with boundary condition, $V_N = p_N r(\infty)$. Optimizing over $t_n$, we obtain the marginal value condition

$$
p_n r'(t_n) = \lambda e^{-\lambda(t_n+\tau)} V_{n+1}.
$$
(M16)

If the rate of detection during casting is a constant $\kappa$, we have $d(t) = 1 - e^{-\kappa t}, r(t) = \kappa \int_0^t e^{-(\kappa+\lambda)s}ds = \frac{\kappa}{\kappa+\lambda} \left( 1 - e^{-(\lambda+\kappa)t} \right)$ and $r'(t) = \kappa e^{-(\kappa+\lambda)t}$. Plugging this expression into (**Equation M16**), we get

$$
t_n = \kappa^{-1} \log \left( \frac{p_n \kappa e^{\lambda\tau}}{\lambda V_{n+1}} \right)
$$
(M17)

Now, let's calculate $V_1$ using Bayesian updates and show that the optimal times exactly correspond to what we have in the previous equation. Denote $\tilde{V}_n(\mathbf{q})$ as the value at patch $n$ for an arbitrary probability vector $\mathbf{q} = (q_1, q_2, \ldots, q_N)$. We now show that $V_1 = \tilde{V}_1(\mathbf{p}_0)$, where $\mathbf{p}_0 = (p_1, p_2, \ldots, p_N)$ is the prior. We have

$$
\tilde{V}_n(\mathbf{q}) = \max_{t_n} \left\{ q_n r(t_n) + e^{-\lambda(t_n+\tau)} \left( 1 - q_n d(t_n) \right) \tilde{V}_{n+1}(\mathbf{q}') \right\},
$$
(M18)

where $\mathbf{q}'$ is the posterior conditional on no detection. The two terms on the right-hand side correspond to the case when the agent finds the target in the patch before $t_n$ (with probability $q_n d(t_n)$) and does not find it (with probability $1 - q_n d(t_n)$), respectively. Given the observation that the target is not found in patch $n$, the posterior probabilities, $\mathbf{q}'$, are obtained using Bayes' rule:

$$
\begin{aligned}
q'_m &= \frac{q_m}{1 - q_n d(t_n)}, \quad \text{for } m \neq n, \\
q'_n &= \frac{q_n(1 - d(t_n))}{1 - q_n d(t_n)}.
\end{aligned}
$$
(M19)

We show that $\tilde{V}_1(\mathbf{p}_0) = V_1$ for $N = 3$. The general case of starting from any patch, prior and number of patches ($N$) follows. Expanding (**Equation M18**) starting from $n = 1$,

$$
\begin{aligned}
\tilde{V}_1(\mathbf{p}_0) &= \max_{t_1,t_2} \left\{ p_1 r(t_1) + e^{-\lambda(t_1+\tau)}(1 - p_1 d(t_1)) \right. \\
&\qquad \left. \times \left( p'_2 r(t_2) + e^{-\lambda(t_2+\tau)}(1 - p'_2 d(t_2))p''_3 r(\infty) \right) \right\},
\end{aligned}
$$
(M20)

where $p'_2$ is obtained from the first Bayesian update and $p''_3$ is obtained after the second Bayesian update. Using (**Equation M19**), we have $p'_2 = p_2/(1 - p_1 d(t_1)), p'_3 = p_3/(1 - p_1 d(t_1))$ and $p''_3 = p'_3/(1 - p'_2 d(t_2)) = p_3/(1 - p_1 d(t_1) - p_2 d(t_2))$.

Since $p''_3(1 - p'_2 d(t_2)) = p'_3$, simplifying (**Equation M20**), we get

$$\begin{aligned}
\tilde{V}_1(\mathbf{p}) &= \max_{t_1,t_2} \left\{ p_1 r(t_1) + e^{-\lambda(t_1+\tau)}(1 - p_1 d(t_1)) \right. \\
&\qquad \times \left. \left( p_2' r(t_2) + e^{-\lambda(t_2+\tau)} p_3' r(\infty) \right) \right\}, \\
&= \max_{t_1,t_2} \left\{ p_1 r(t_1) + e^{-\lambda(t_1+\tau)} \left( p_2 r(t_2) + e^{-\lambda(t_2+\tau)} p_3 r(\infty) \right) \right\},
\end{aligned} \tag{M21}$$

where $p_2' = p_2/(1 - p_1 d(t_1)), p_3' = p_3/(1 - p_1 d(t_1))$ are used in the second step. This equation exactly corresponds to (*Equation M14*). The upshot is that the normalization factors from the Bayesian updates go through the parenthesis and cancel out. However, optimizing for $t_n$ directly using (*Equation M18*) is difficult due to the dependence of $\mathbf{q}'$ on $t_n$.

# Acknowledgements

GR was partially supported by the NSF-Simons Center for Mathematical & Statistical Analysis of Biology at Harvard (award number #1764269) and the Harvard Quantitative Biology Initiative. This work received support from: the European Research Council (ERC) under the European Union's Horizon 2020 research and innovation programme (grant agreement No. 101002724 RIDING); the Air Force Office of Scientific Research under award number FA8655-20-1-7028; the National Institutes of Health (NIH) under award number R01DC018789. The authors are grateful to the OPAL infrastructure from Université Côte d'Azur and the Université Côte d'Azur's Center for High-Performance Computing for providing resources and support. NR is thankful for the support of Instituto Nazionale di Fisica Nucleare (INFN) Scientific Initiative SFT. This research was initiated at the Kavli Institute for Theoretical Physics supported in part by NSF Grant No. PHY-1748958 and the Gordon and Betty Moore Foundation Grant No. 2919.02.

# Additional information

## Competing interests

Agnese Seminara: Reviewing editor, *eLife*. The other authors declare that no competing interests exist.

## Funding

| Funder | Grant reference number | Author |
|---|---|---|
| National Science Foundation | 1764269 | Gautam Reddy |
| European Research Council | 101002724 | Agnese Seminara |
| Air Force Office of Scientific Research | FA8655-20-1-7028 | Agnese Seminara |
| NIH Office of the Director | R01DC018789 | Agnese Seminara |
| National Science Foundation | PHY-1748958 | Massimo Vergassola |
| Gordon and Betty Moore Foundation | 2919.02 | Gautam Reddy |

The funders had no role in study design, data collection and interpretation, or the decision to submit the work for publication.

## Author contributions

Nicola Rigolli, Gautam Reddy, Agnese Seminara, Massimo Vergassola, Conceptualization, Investigation, Methodology, Writing – original draft, Writing – review and editing

## Author ORCIDs

Nicola Rigolli ⓘ http://orcid.org/0000-0002-0734-2105
Gautam Reddy ⓘ http://orcid.org/0000-0002-1276-9613
Agnese Seminara ⓘ http://orcid.org/0000-0001-5633-8180

Massimo Vergassola ⬛ http://orcid.org/0000-0002-7212-8244

**Decision letter and Author response**
Decision letter https://doi.org/10.7554/eLife.76989.sa1
Author response https://doi.org/10.7554/eLife.76989.sa2

## Additional files

### Supplementary files
• Transparent reporting form
• Supplementary file 1. Data and scripts to generate *Figures 3–6*.

### Data availability
All data generated or analyzed during this study are included in the manuscript and supporting file. The dataset with the simulation results has been made public at https://zenodo.org/record/6538177#. Yqrl_5BByJE.

The following dataset was generated:

| Author(s) | Year | Dataset title | Dataset URL | Database and Identifier |
|---|---|---|---|---|
| Rigolli N, Reddy G, Seminara A, Vergassola M | 2022 | Alternation emerges as a multi-modal strategy for turbulent odor navigation - Dataset | https://doi.org/10.5281/zenodo.6538177 | Zenodo, 10.5281/zenodo.6538177 |

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
