## [Editor Report]

This work demonstrates how animals can combine different kinds of sensing actions to improve the accuracy of finding the sources of olfactory signals. Previous work have provided an explanation of the casting and surging behaviors used to find the plume in order to navigate towards the sources. This work adds "alternation" - sniffing in the air far from the ground - that animals can do by rearing on hind legs. The authors show that alternation occurs more frequently far away from the source and that this can be explained by the marginal value theory.

---

## [Decision Letter]

**Decision letter after peer review:**

Thank you for submitting your article "Alternation emerges as a multi-modal strategy for turbulent odor navigation" for consideration by *eLife*. Your article has been reviewed by 2 peer reviewers, one of whom is a member of our Board of Reviewing Editors, and the evaluation has been overseen by Ronald Calabrese as the Senior Editor. The reviewers have opted to remain anonymous.

Essential revisions:

It is important to add discussions of implementation constraints associated with memory requirements and assumptions about wind parameters and movement of the sources as discussed in detail by Reviewer 1.

Reviewer 2 stresses the importance of making the simulation dataset easily accessible to readers.

*Reviewer #2 (Recommendations for the authors):*

Please make the dataset with the simulation results available. I did not see it in the zip file.

Please provide color for velocity fluctuations in Figure 1.

Please clarify the statement on line 134 that the model lacks the appropriate spatiotemporal correlations.

Line 139 and ff: there are missing words and text. For example, there is (1) but no subsequent points. In any case, please add more here about how hyper-planes are constructed.

The possible set of actions is described in line 111ff but the rate of sniffing is not described as one of the variables there. Later (line 172ff), the authors discuss the frequency of sniffing as a function of distance from the source. Therefore, this frequency should be explicitly included in the initial setup of the model.

Line 178 double word: "the the".

---

## [Author Response]

Reviewer #2 (Recommendations for the authors):Please make the dataset with the simulation results available. I did not see it in the zip file.

The dataset has been made public at https://zenodo.org/record/6538177#.Yqrl_5BByJE

Please provide color for velocity fluctuations in Figure 1.

The flow is an illustration and colors have been chosen for visual purposes. We prefer then to keep things as they are and to take the above remark into account by changing the caption so as to make it clear that it is an illustration that has the purpose of providing an artistic view of the dynamics.

Please clarify the statement on line 134 that the model lacks the appropriate spatiotemporal correlations.

The sentence “The model is partially inaccurate as the average detection rate does match the simulated odor plumes but the model of the agent lacks the appropriate spatiotemporal correlations because detections are independent in the Poissonian model and they are not in the real flow.” (Lines 133-135) has been added for clarification.

Line 139 and ff: there are missing words and text. For example, there is (1) but no subsequent points. In any case, please add more here about how hyper-planes are constructed.

We have modified the text to describe in some more detail the idea for the construction of the hyperplanes without consulting the Methods. The relevant lines are 138-143.

The possible set of actions is described in line 111ff but the rate of sniffing is not described as one of the variables there. Later (line 172ff), the authors discuss the frequency of sniffing as a function of distance from the source. Therefore, this frequency should be explicitly included in the initial setup of the model.

The frequency of sniffing is fixed throughout the search and is not modified as a function of distance to the source. What we meant by “rate of sniffing the air” is the fraction of time spent in action (vi) among the six possible alternative actions offered to the searcher ((i-iv) move to one of the four neighboring locations while sniffing the ground, (v) stay at the same location and sniff the ground or (vi) stay at the same location and sniff the air). The sentence at line 175 has been modified to clarify the possible ambiguity as: “… and measure the rate of sniffing the air (fraction of time spent in the action (vi) -- staying at the same location and sniffing the air -- among the six possible alternative actions offered to the searcher) for both of these behaviors.”

Line 178 double word: "the the".

The typographical error has been corrected.